# Low-Rank Successor Representations Capture Human-Like Generalization

**Eva Yi Xie**[1]    **Nathaniel Daw**[1]*    **Benjamin Eysenbach**[2]*
[1] Princeton Neuroscience Institute, Princeton University, NJ, USA
[2] Department of Computer Science, Princeton University, NJ, USA
{evayixie, ndaw, eysenbach}@princeton.edu

## Abstract

Intelligent behavior is hypothesized to hinge on predictive maps of future states. The successor representation (SR) formalizes such maps and has been influential in reinforcement learning and cognitive neuroscience. The SR is typically represented in a tabular form, where each entry stores the probability of transitioning from one state to another, but this approach does not generalize to states unseen during training. We thus study the *function approximation* of the SR, asking whether it yields more efficient solutions to navigation tasks and captures human-like behavioral patterns. Potentially, the same representational structure that supports efficient navigation may also shape systematic biases in behavior. To learn such approximations, we use a contrastive reinforcement learning objective. In gridworlds, we find that function-approximated SRs generalize to novel states and produce smoother representations; this includes successfully navigating out-of-distribution state-goal pairs and generating shorter paths on tasks solvable by tabular SRs. In high-dimensional graph learning benchmarks, the function-approximated SRs are low-rank and resemble human behavioral patterns across graph topologies, whereas tabular SRs would lack such inductive bias. Taken together, our results position function-approximated SRs as a practical framework for probing the representational structure that underlies flexible and efficient planning in both biological and artificial systems.

## 1 Introduction

Intelligent behavior is hypothesized to hinge on predictive maps of how states unfold under action [Stachenfeld et al., 2017]. The successor representation (SR) [Dayan, 1993] formalizes such maps by organizing states according to their expected discounted future occupancies, and has become a shared idea in deep [Barreto et al., 2017, Eysenbach et al., 2022] and human [Garvert et al., 2017, Momennejad et al., 2017] reinforcement learning.

In tabular settings, the SR can be estimated directly by counting state-to-state transitions, yielding a full matrix of future state visitation frequencies [Dayan, 1993, Russek et al., 2017]. Such matrices are typically high-rank, faithfully preserving fine distinctions between states. However, such tabular estimates break down in large or continuous environments. In these settings, the states cannot simply be enumerated, and tabular estimates for unseen states would be undefined or remain at arbitrary initialization values. To generalize across these settings, agents must rely on function approximation [Gershman, 2018], which replaces a lookup table with a parameterized mapping (*e.g.*, a neural network) that estimates predictive relationships from observed samples. Ideally, such representations would also allow agents to combine experience across distinct episodes by recognizing structural

---

*These two senior authors contributed equally.

similarities in otherwise different observations, such as when walking the same path in the day and later at night. This raises a central question for both biological and artificial learning agents: do function approximations of predictive representations support more efficient planning and broader generalization? Yet it is often unclear what kinds of features are learned under function approximation, or how they relate to the structure of the underlying environment. As an analytical tool, we return to the tabular setting and study low-rank parameterizations.

**Related work and our contribution.** Human graph-learning studies reveal SR-like regularities in reaction times across various graph topologies, suggesting that people acquire predictive maps of transitions [Kahn et al., 2018]. In parallel, recent theory connects contrastive reinforcement learning objectives to SR-like predictive distances under function approximation [Eysenbach et al., 2022]. Building on these threads, we cast SR learning as a *low-rank* function approximation trained with a contrastive RL objective. We show that the resulting SRs generalize to unseen states and yield shorter plans than tabular SRs in grid-world simulations. Additionally, we demonstrate that these SRs are low-rank and capture human reaction-time patterns reported in Kahn et al. [2018].

## 2 Methods

**Contrastive reinforcement learning** We adopt a contrastive objective to learn function-approximated SRs, following Eysenbach et al. [2022]. On a finite state set $\mathcal{S}$, given trajectories $(s_0, \ldots, s_T)$, we learn low-dimensional encoders $h_\theta, g_\theta : \mathcal{S} \to \mathbb{R}^d$ with $d \ll |\mathcal{S}|$. We define a *symmetric* distance

$$d_\theta(s, s') = \big\| [h_\theta(s), g_\theta(s)] - [h_\theta(s'), g_\theta(s')] \big\|_2, \quad \text{and score} \quad f_\theta(s, s_g) = -d_\theta(s, s_g).$$

Each minibatch (size $m$) samples $m$ short windows; for each window we choose an *anchor* $s_t$ and a *positive* sample $s_{t+\tau}$ from the *same* trajectory, with $\tau \sim p(\tau) \propto \gamma^\tau$, where $\gamma \in (0, 1)$. The remaining $m-1$ targets in the batch act as *negatives*. The forward InfoNCE loss is

$$\mathcal{L}_{\text{forward}} = -\mathbb{E}\left[ \log \frac{\exp\{f_\theta(s_t, s_{t+\tau})\}}{\sum_{x \in \mathcal{N}_t} \exp\{f_\theta(s_t, x)\}} \right],$$

where $\mathcal{N}_t$ contains the one positive and $m-1$ negatives for anchor $s_t$. We train $\theta$ for a fixed number of gradient steps.

**Planning with the learned distance** In grid worlds, states are grids, and a reinforcement learning agent moves to adjacent grids through actions (respecting walls). Given a goal $g \in \mathcal{S}$, we perform one-step look-ahead using $d_\theta(s, s')$: from $s$, we evaluate all neighbors $s'$ and choose the next state according to a soft-greedy policy. Specifically, the probability of selecting $s'$ is $p(s' \mid s, g) \propto \exp\big[ -d_\theta(s', g)/\tau \big]$, where $\tau$ is the temperature parameter. Lower distances are exponentially more likely, with $\tau \to 0$ recovering deterministic greedy selection and large $\tau$ yielding nearly uniform choices. We fix $\tau = 1$, giving a moderately sharp preference for the closest neighbor while allowing occasional stochasticity. Success is measured as the fraction of start–goal pairs that reach $g$ within a fixed step limit.

**Participation ratio** Given a distance matrix $D \in \mathbb{R}^{n \times n}$ with $D_{ij} = d_\theta(s_i, s_j)$, we compute its eigenspectrum through multidimensional scaling [Wickelmaier, 2003]. Let $\{\lambda_i\}$ be the positive eigenvalues. The effective dimensionality is then computed as the participation ratio [Kramer and MacKinnon, 1993, Gao et al., 2017, Recanatesi et al., 2022], $D_{\text{eff}} = \left( \sum_i \lambda_i \right)^2 / \sum_i \lambda_i^2$.

## 3 Results

### 3.1 Function-approximated SR generalizes and shortens paths

We study function-approximated SR using a contrastive reinforcement learning (CRL) objective [Eysenbach et al., 2022]. The CRL objective estimates a predictive distance under function approximation by shaping representations to separate likely from unlikely futures. As a corollary of Theorem 4.1 in Eysenbach et al. [2022], it follows that the optimal CRL critic recovers a form of the SR under

standard assumptions. We hypothesize that function-approximated SRs generalize to unseen states and capture smoother relational structure, whereas tabular SRs provide meaningful representations only for state pairs observed during training, leading to limited task performance.

To test this hypothesis, we compare tabular and function-approximation approaches in a simple grid world navigation task (Fig. 1A). In this task, agents learn from 300 trajectories of length 5 sampled from a random policy, and must then navigate between randomly chosen start-goal pairs. We observe that the function approximation approach generalizes to out-of-distribution start–goal pairs ($\geq 10$ steps apart) and sustains higher success rates than the tabular SR baseline and a random baseline (Fig. 1B). Moreover, conditioned on tasks that tabular SRs can solve (success rate $\geq 0.8$), the function-approximated SR finds shorter paths, as reflected by the right-shifted distribution of normalized improvement (Fig. 1C).

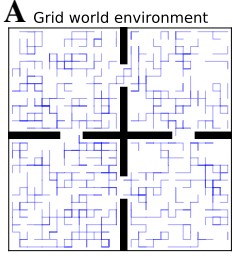
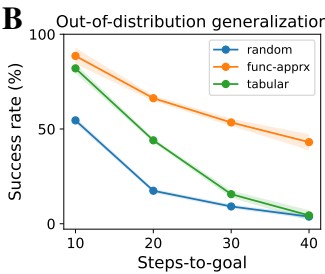
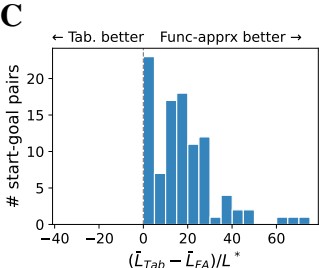

Figure 1: **Function-approximated SR generalizes and shortens paths. (A)** Visualization of the grid world with the 300 length-5 randomly sampled training trajectories (blue), respecting the walls (black). **(B)** Planning performance by steps-to-goal after training on 300 length-5 trajectories from a random policy. The evaluated start-goal pairs are out-of-distribution. Here, the function-approximated SR (trained with an embedding network representation dimension of 1) sustains higher success than the tabular SR and the random baseline, especially at longer distances. We show the mean over 5 seeds with standard error. **(C)** Histogram of the normalized path-length improvement $(\bar{L}_{\text{Tab}} - \bar{L}_{\text{FA}})/L^*$, where $\bar{L}$ is the mean path length over 5 rollouts and $L^*$ is the optimal shortest-path length. We include only cases for which the tabular SR succeeds (success rate $\geq 0.8$); positive values indicate the function-approximated (FA) SR finds shorter paths. We provide detailed visualization in Appendix A.

### 3.2 Function-approximated SR is low-rank and resembles human behavior on graphical tasks

We next evaluate function-approximated SRs on graph-learning benchmarks commonly used to probe human successor representations [Garvert et al., 2017, Kahn et al., 2018, 2025]. In these tasks, human participants perform a serial reaction-time paradigm: each node of an underlying graph (*e.g.*, modular or lattice; Fig. 2A) is mapped to a key-press sequence on a keyboard. Participants are then presented with node-corresponding stimuli generated by a random walk over the graph. Although participants are not told that a graph underlies the sequence, their reaction times (RTs) reveal a learned predictive map of transitions [Kahn et al., 2018]. We ask whether SRs learned with function approximation, which provide benefits in navigation tasks, exhibit low-rank structure and reproduce qualitative behavioral signatures observed in humans on high-dimensional graph-learning tasks.

We found that human RT ordering across graph types (reported in Kahn et al. [2018], Fig. 2C) aligns with our analysis of effective graph dimensionality (Fig. 2B) using participation ratio [Kramer and MacKinnon, 1993, Gao et al., 2017, Recanatesi et al., 2022]. Specifically, modular graphs have the lowest dimensionality and also elicit the fastest RTs, followed by lattice and random graphs (Fig. 2C). This correspondence suggests that differences in effective dimensionality may help explain the relative difficulty of graph types observed in human behavior, though confirming this link will require future investigation. On the same benchmarks, function-approximated SRs saturate at a low effective dimensionality, namely *low-rank*, below that of the graphs themselves (Fig. 2B). This low-rank property holds even when the embedding network used for function approximation has a high representational dimension (Fig. 2D). These results suggest that effective dimensionality offers a candidate explanation for human RT differences, and that function-approximated SRs exhibit a bias toward lower-dimensional representations.

Further, we found the low-rank SRs qualitatively capture graph effects observed in human behavior [Kahn et al., 2018]. For example, humans respond faster to within-cluster than to across-cluster transitions in modular graphs, and low-rank SRs show an analogous pattern by assigning shorter distances within clusters (cluster effect, Fig. 2E) after being trained on trajectories drawn from a cluster graph. Secondly, humans exhibit slower responses to transitions that were never experienced during training [Kahn et al., 2018]. We similarly trained on trajectories drawn from a 15-node modular, lattice, or random graph and then evaluated novel edges of a 15-node fully connected graph. We found the low-rank SR predicts larger distances for edges unseen during training (novel-edge effect, Fig. 2F). Notably, a tabular SR would assign arbitrary default values to unseen edges, since they are absent from its training data. Finally, the low-rank SR captures the relative reaction time ordering across graph types seen in human RTs (graph effect, Fig. 2C). In our analysis, we trained the CRL agent on one graph type and tested on one of the two remaining graph types. We then measured the differences in average predicted distances between the graph types. This simulates the human experiment [Kahn et al., 2018], where participants were exposed to one graph during training and a different graph during testing, and RT differences were taken across orders. Consistent with the human data (Fig. 2C), we found that predicted distances for random versus modular graphs were largest (green), followed by lattice versus modular (orange). The difference of random versus lattice was near zero (blue), mirroring the non-significant RT differences in humans for that pair (Fig. 2C).

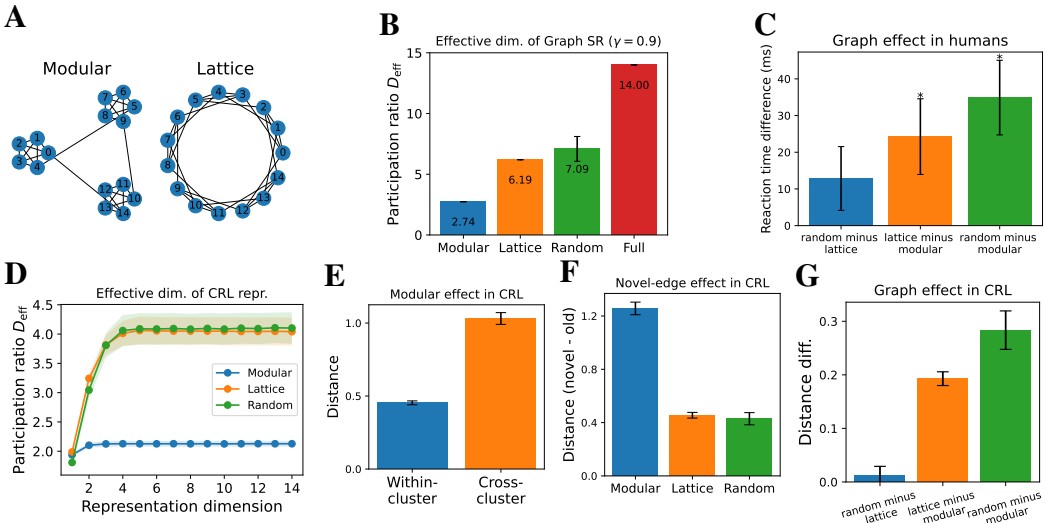

Figure 2: **Low-rank SR captures human graph effects.** (**A**) Schematics of the task graphs (modular, lattice) used in the human experiments of Kahn et al. [2018]. (**B**) Effective dimensionality ($D_{\text{eff}}$, participation ratio) of each graph type in Kahn et al. [2018], computed from its SR-based distance with discount $\gamma=0.9$; the modular graph has the lowest $D_{\text{eff}}$. (**C**) Human reaction time ordering follows Modular $<$ Lattice $\approx$ Random (lower is easier), consistent with the dimensionality differences in (B). (**D**) Effective dimensionality ($D_{\text{eff}}$, participation ratio) of the distance induced by the function-approximated SR quickly saturates at a low rank even as the embedding network representation dimension increases, across graph types. (**E**) *Cluster effect:* The low-rank SR assigns shorter distances to within-cluster than across-cluster transitions on modular graphs, matching reported human behavior [Kahn et al., 2018]. (**F**) *Novel-edge effect:* After training on Modular/Lattice/Random graphs, the low-rank SR predicts larger distances for unseen (novel) edges when evaluated on a fully connected test graph, mirroring slower human RTs; tabular SRs would instead assign arbitrary default values to such edges. (**G**) *Graph effect:* The low-rank SR captures the relative reaction time pattern of panel C. We simulate the representation dimension of 1 in CRL to generate panels E, F, and G, and plot the mean and standard error across 5 seeds for panels D–G.

## 4 Discussion

Our results position function-approximated SRs as a tractable framework to study predictive maps and their dimensionality in both biological and artificial systems. A key limitation is that our comparison to human data was qualitative. Our immediate next step is to directly fit human reaction times [Kahn

et al., 2018] with function-approximated SRs. Moreover, our tasks were restricted to simple grid worlds, and it is important to extend our analyses to richer, more naturalistic environments. Finally, future studies should consider involving neural data, *e.g.*, testing whether hippocampal-entorhinal representations exhibit similar low-rank predictive structure [Kahn et al., 2025, Garvert et al., 2017].

## 5    Acknowledgments

We gratefully acknowledge financial support from the Princeton Laboratory for Artificial Intelligence's Natural and Artificial Minds initiative.

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

# A    Visualizations of random roll-outs in grid world

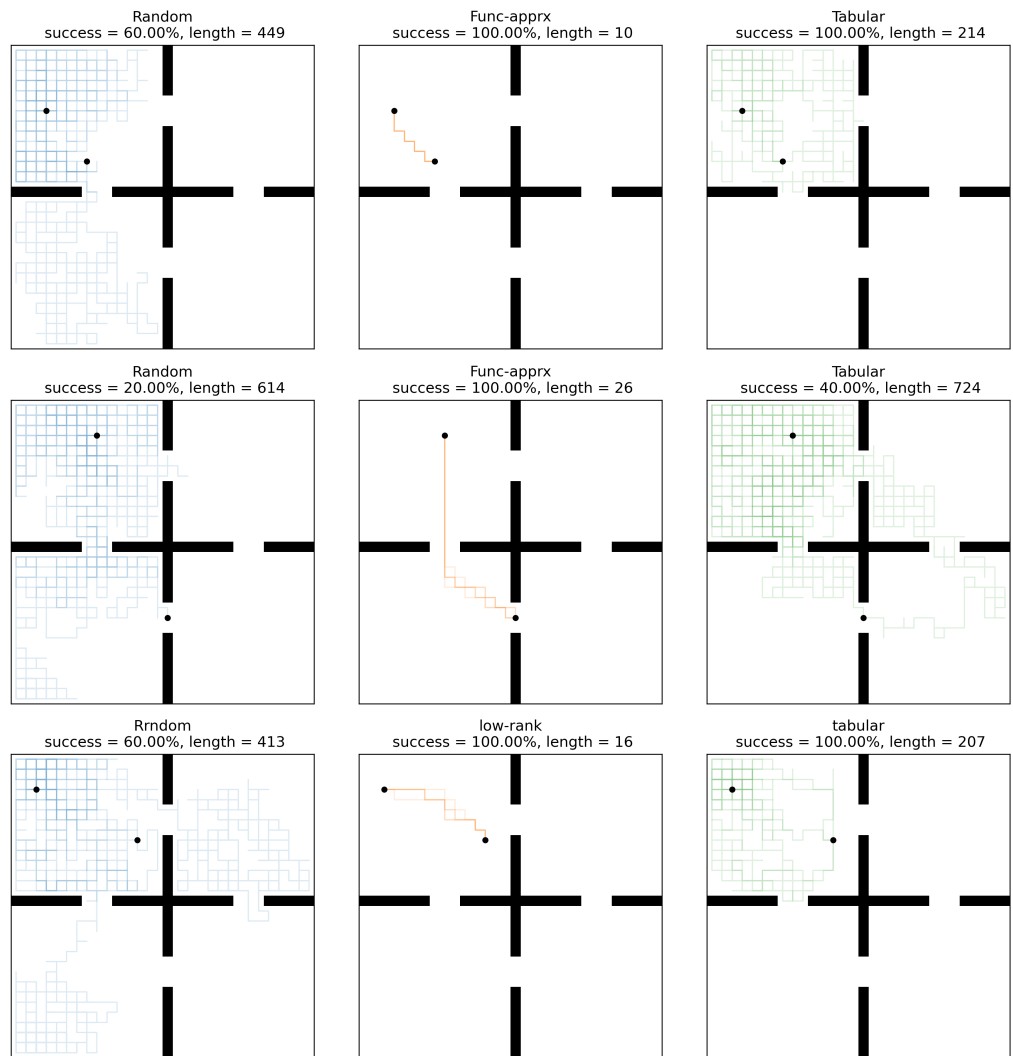

Figure 3: **Example roll-outs in a grid world with plus-shaped walls.** We visualize agent trajectories under a soft-greedy policy ($\tau = 1$) using three distance representations: uniformly random (left, blue), function-approximated SR (middle, orange), and tabular SR (right, green). Dots mark randomly sampled start-goal pairs unseen during training; black bars denote walls. Panel titles report success rate (%) and mean path length, averaged over 5 roll-outs per start-goal pair.

