# OpenReview forum: "Low-Rank Successor Representations Capture Human-Like Generalization"
_NeurIPS.cc/2025/Workshop/UniReps — UniReps2025_

### Official Review · Reviewer_9a3B · 2025-09-15
**Promising approach, more discussion regarding complex, real-worlds scenarios would strengthen the work**

**Confidence:** 4

**Review:**

Summary

This work explores function-approximated successor representations learned via contrastive RL, contrasting them with traditional tabular approaches. The authors show that these SRs can generalize to novel states in grid navigation and display low-rank structure that mirrors human behavioral patterns in graph learning tasks. The main contribution is demonstrating that learned representations maintain lower effective dimensionality than underlying graphs while reproducing human reaction time effects like cluster preferences, novel-edge penalties, and cross-graph ordering patterns.

Strengths

1. The work addresses a fundamental limitation of tabular SRs - their inability to generalize to unseen states. The motivation for studying function approximation is clear and well-articulated, particularly the connection between efficient planning and human-like behavioral biases.
2. Linking contrastive RL with cognitive neuroscience findings is compelling, especially with low-rank structure correspondence with human graph learning behavior.
3.Using participation ratio as a measure of effective dimensionality is good way to characterize the low-rank structure and connects nicely to human behavioral data.
4. The experiments are well-designed to test specific hypotheses. The out-of-distribution generalization tests in gridworlds and comparison to human behavioral effects seem well chosen to validate key claims.

Weaknesses

1. The alignment with human data, while compelling, remains qualitative. The authors acknowledge this limitation and propose quantitative fitting as future work, but some discussion of potential challenges or preliminary findings would strengthen the submission.
2. Key implementation details are sparse - architecture of embedding functions, training hyperparameters, and sensitivity analyses are not provided. While understandable for an extended abstact, some discussion of stability and robustness would helpful.
3. The authors acknowledge this but its still a major limitations. Without actual RL fits, we cant tell if the model actually captures human behavior or just show similar trends.
4. Experiments are restricted to relatively simple environments and basic graph structure. Would be valuable to discuss how the approach might scale to more complex, realistic environments.

**Score:**

4

**Topic Fit:**

3

---

### Official Review · Reviewer_4ks2 · 2025-09-15
**Interesting Insights to Contrastive RL-based functional approximation of successor representations.**

**Confidence:** 2

**Review:**

Summary: This paper introduces a new approach to successor representation for predictive maps of future states. The core idea is to use a contrastive reinforcement learning objective to learn low-dimensional encoders of states. The authors provide some interesting insights to successor representation-based approaches demonstrating that function-approximated successor representations generalize more effectively to unseen/out-of-distribution environments compared to tabular methods.

Clarity: The writing could be improved, as certain sections are difficult to follow. In particular, the methodology section would benefit from a clearer, more structured presentation, and the experimental section is quite densely written, making it challenging to parse key results.

Novelty: The paper proposes a novel approach that offers valuable insights and has the potential to spark further discussion on successor representations in reinforcement learning.

**Score:**

3

**Topic Fit:**

2

---

### Official Review · Reviewer_Y7fF · 2025-09-15
**SR low rank function approximation + human response time**

**Confidence:** 3

**Review:**

This Extended abstract adresses the topic of function approximation when approximating the next state of a process. They provide a formalization of the framework, a set of experiments comparing function approximation to tabular approaches, and make interesting hypothesis about links to human processing. They show that tabular approaches have (expectedly) weaker generalization to unseen sequences.
When comparing to human data, they show that there are faster human responses when successor representations can be expressed in Low rank.
I am neither an expert on the litterature for human successor representations, or reinforcement learning, and cannot guarantee that no major work in the domain has been overlooked. I will defer to other reviewers on this point.
Nonetheless, the topic seems relevant to the workshop.

I would raise a few metholodogical questions. As they will not change in a major way my evaluation of the paper, they should be treated as potential discussion topics for further work and not required answers for acceptance :

 - Is there a hypothesis that the loss used (Contrastive Reinforcement Learning) in any way relates to the human learning process? There has been some debate whether the link between correlation of representations in the brain language region and LLMs was in anyway related to a next token prediction objective in humans. While correlation is strong, conclusions on relation between objective functions remain weak

- Low rank modules attempt to regroup the main directions of variance in a high dimensional space. Is there an intuitive understanding of why human response time would be shorter in the main directions of variance?

**Score:**

4

**Topic Fit:**

3